# Effect of Age on Heart Rate Variability in Patients with Mitral Valve Prolapse: An Observational Study

**DOI:** 10.3390/jcm12010165

**Published:** 2022-12-25

**Authors:** Jau-Kang Huang, Shiang-Yun Huang, Chih-hsien Lee, Ing-Fang Yang, Ten-Fang Yang, Yun-Ming Wang

**Affiliations:** 1National Yang Ming Chiao Tung University, Hsin-Chu 300093, Taiwan; 2Shin Kong Wu Ho-Su Memorial Hospital, Taipei 111045, Taiwan; 3Tung’s Taichung Metrohabor Hospital, Taichung 435403, Taiwan; 4Jen Chi General Hospital, Taipei 108016, Taiwan; 5Taipei Medical University and Hospital, Taipei 110301, Taiwan

**Keywords:** heart rate variability, arrhythmia, mitral valve prolapse, Holter, time domain, frequency the domain, RR interval, autonomic nervous system

## Abstract

Age is an important determinant of heart rate variability (HRV) in healthy individuals. The incidence of arrhythmia is high in patients with mitral valve prolapse (MVP). However, the correlation of HRV in patients with MVP in different age groups is not well established. We presumed that increasing age would be prospectively associated with declining HRV measurement in MVP. Sixty patients with MVP and 120 control individuals were included and underwent 24 h HRV analysis. No significant difference was found in all parameters calculated in the time domain or in the frequency domain between the two groups. However, as patients’ age increased, a significant time domain (SDNN, RMSSD, NN50, and pNN50) decline was found in the MVP group, but not in the control group. This suggests that patients with MVP may have autonomic nervous system involvement that increases the risk of arrhythmia and heart disease with increasing age.

## 1. Introduction

In most cases, mitral valve prolapse (MVP) is a benign disease and seldom causes cardiac events, such as arrhythmia and sudden cardiac death [1,2,3]. MVP is the most common valve abnormality and is estimated to affect 2–3% of the population [4,5,6]. According to previous studies, a subset of patients with symptomatic MVP have associated autonomic dysfunction, dominant sympathetic nerve involvement, and attenuated parasympathetic nerve tone [6,7,8].

Evaluating heart rate variability (HRV) is a noninvasive and simple method of studying autonomic function [9,10]. Several studies have shown that reduced HRV is significantly associated with cardiovascular diseases and sudden deaths [11,12,13]. Abnormal HRV is also found in many diseases, such as diabetes and central nervous system diseases [14,15,16,17,18]. Generally, HRV is influenced by several factors such as chemical, hormonal, and neural modulations, circadian changes, exercise, emotions, posture, and preload. The adaptation of the heart rate to changing factors is attributed to the activity of various regulatory subsystems and results in complex linear and nonlinear time behaviors, which change with age and pathological conditions, such as MVP [19,20,21,22,23,24]. Furthermore, age has a strong influence on the heart and HRV [25,26,27,28,29,30,31]. However, the correlation of HRV in patients with MVP in different age groups is not well established. HRV is affected by several factors, including MVP. We hypothesize that the effect of MVP on HRV is independent of age. Thus, this study aimed to investigate the correlation of HRV on different age groups of patients with MVP.

## 2. Materials and Methods

### 2.1. Ethical Information

Approval for this study was granted by the Ethics Committee of the New Taipei City Hospital (REC NTPC No. 108003-E) for data protection and privacy. All participants were informed about the study and gave their written consent. All the study procedures were performed following the principles described in the Declaration of Helsinki.

### 2.2. Study Population

A total of 180 patients with palpitation and chest pain from an outpatient clinic were included. Patients with smoking and coffee consumption were asked to avoid caffeinated drinks and smoking during the study. They were not taking any medication. They had no significant physical examination, chest X-ray, or electrocardiography findings. All study participants underwent a standard two-dimensional transthoracic echocardiography using a commercially available system (ClearVue 350, Phillips Medical Products, Bothell, WA, USA) with a 2.5 MHz transducer. Furthermore, MVP was diagnosed using the current two-dimensional echocardiographic criteria [32,33], as leaf displacement of >2 mm beyond the mitral annulus in a parasternal or apical three-chamber long-axis view. After applying the exclusion criteria (i.e., mitral valve regurgitation ≥ moderate degree, mitral annular disjunction, atrial fibrillation, end-stage renal disease, hypertension, diabetes, hyperthyroidism, coronary artery disease, and sleep obstructive apnea), 60 patients with MVP (MVP group) and 120 controls (control group) remained.

### 2.3. HRV Evaluation

Time series of heart rates consisting of beat-to-beat intervals (RR intervals) was extracted from 24 h ambulatory Holter electrocardiography recordings (DR200/HE, Holter and Event Recorder, North East Monitoring Inc., Maynard, MA, USA). Data from the recorder were analyzed using a Holter analyzer (HE/LX Holter Analysis Software).

### 2.4. Time Domain (TD)

Based on the Task Force of the European Society of Cardiology and the North American Society of Pacing and Electrophysiology [34], the following standard HRV indices from the TD were calculated:SDNN (ms): standard deviation of the NN interval time series.RMSSD (ms): square root of the mean squared differences of successive NN intervals.NN50: number of pairs of successive NN (R-R) intervals that differ by >50 ms.pNN50 (arbitrary units): proportion derived by dividing the number of interval differences of successive NN intervals > 50 ms by the total number of NN intervals.

### 2.5. Frequency Domain (FD)

Equal to the TD indices, the following standardized HRV indices from the Task Force Guideline of the FD were extracted:Total power (ms^2^): total power of the power spectral density in the range of frequencies between 0 and 0.4 Hz.VLF (ms^2^; very low frequency): a band of power spectrum range between 0.0033 and 0.04 Hz.LF (ms^2^): power in the “low”-frequency band (0.04–0.15 Hz).HF (ms^2^): power in the “high”-frequency band (0.15–0.4 Hz).LF/HF (arbitrary units): ratio between LF and HF.

### 2.6. Statistical Analysis

The characteristics of the MVP group and the control group were compared using an independent sample *t*-test for continuous variables or Fisher’s exact test for categorical variables. The variation in the parameters calculated in the TD (SDNN, RMSSD, NN50, and pNN50) and in parameters calculated in the FD (total power, VLF, LF, HF, and LF/HF) was large. The Kolmogorov–Smirnov normality tests showed that none of these HRV parameters met the normal distribution assumptions. Therefore, all analyses regarding HRV parameters were nonparametric. The parameters calculated in the TD (or in the FD) between the MVP and control groups were compared using the Mann–Whitney U-test. We created an additional propensity score-matched cohort to avoid potential confounding because the mean age of the MVP group was much younger than that of the control group. The listed basic demographics in Table 1 were used to calculate the propensity score from a multivariable logistic regression model. The difference in HRV parameters between the MVP and control groups was further tested using multivariable quantile regression using the whole cohort (*n* = 180). The characteristics that showed borderline difference (*p* < 0.25) between the two groups were adjusted, including age, sex, and smoking. The desired quantile was the 50th percentile, and the standard error was calculated under the assumption of independent and identically distributed errors. Finally, the relationship between age and HRV parameters was explored using Spearman’s rank correlation. A two-sided *p*-value of <0.05 was considered significant. Statistical analyses were conducted using IBM SPSS version 26 (IBM Corp., Armonk, NY, USA).

## 3. Results

### 3.1. Patient Characteristics

The mean age of the MVP group was 39 (standard deviation (SD) 15.4) years, with predominance of female patients (76.7%) (left panel in Table 1). Eleven (18.3%) patients smoked, and 10 (16.7%) were coffee drinkers. Six (10%) patients had insomnia and had a history of syncope. The MVP group was significantly younger than the control group (39 years vs. 48.9 years, *p* < 0.001). No significant difference was found in the remaining characteristics between the groups. Each group had 52 participants after propensity score matching, and none of the characteristics significantly differed between the two groups (right panel in Table 1).

### 3.2. HRV Parameters between the MVP and Control Groups

No significant difference was noted in all HRV parameters between the two groups in either the whole cohort (left panel in Table 2) or the propensity score-matched cohort (right panel in Table 2). Multivariable quantile regression revealed no associations between MVP and HRV parameters after adjusting for age, sex, and smoking in the whole cohort (Table 3). The effect of age was significant on the following HRV parameters: SDNN, RMSSD, NN50, PNN50, mean RR interval, LF, and total RR interval (Appendix A).

### 3.3. Relationship between Age and HRV Parameters

In the MVP group, patients’ age significantly correlated with SDNN (ρ = −0.34, *p* = 0.008), RMSSD (ρ = −0.39, *p* = 0.002), NN50 (ρ = −0.47, *p* < 0.001), and PNN50 (ρ = −0.46, *p* < 0.001) values (Table 4). However, no significant relationships between age and RR interval and FD parameters were found in the MVP group. By contrast, the correlation between age and SDNN (ρ = −0.18), RMSSD (ρ = −0.14), and NN50 (ρ = −0.32) was apparently weaker in the control group than in the MVP group (Figure 1a–c). More interestingly, the correlation between age and RR interval and FD parameters was apparently stronger in the control group than in the MVP group (Figure 2a–d).

## 4. Discussion

Several studies have shown that patients with MVP have a predominant sympathetic effect [21,22,23,24,25,26] and reduced parasympathetic tone [35]. In adults with MVP, some HRV data showed no difference in the autonomic nerve tests, and the autonomic nerve abnormalities may be caused by symptoms rather than MVP itself, so information still has some inconsistencies [36,37] but implies a marked deviation in the tone of the autonomic nervous system dominated by the sympathetic effect, especially in symptomatic cases [36]. However, the results of our study showed no significant difference between the HRV recorded from the MVP and control groups. This may imply that some confounding factors influence our results such as age distribution and case number.

Numerous studies have shown the effect of factors such as age, sex, and some physiological response on cardiac autonomic function [38,39] and have also presented that SDNN, SDANN, and RMSSD negatively correlate with age [38,40]. This suggests that cardiac autonomic responses are attenuated with age, especially the modulation of parasympathetic nerves. Our study also gave the same results. Furthermore, the MVP groups had a more significant negative correlation with age than the control subjects did. Such results reveal that the MVP population had a more significant effect on the autonomic nervous system as age increased, making HRV more significantly negatively correlated.

A study also showed that sympathetic tone predominates, while parasympathetic tone also diminishes with age [38]. Although parasympathetic activity decreases with age, some endocrine factors, including thyroxine, reproductive hormones, and renin–angiotensin effects on sympathetic tone decrease with age, and circadian hormone and temperature rhythms decrease with age [29,38,41,42]. This indicates that many factors can affect the HRV of healthy individuals or those with some diseases. In our study, a significantly negative correlation with HRV TD parameters with increasing age was found in patients with MVP. However, no significant relationships between age and RR interval and FD parameters were found in the MVP group. By contrast, the correlation between age and TD parameters was apparently weaker in the control group than in the MVP group. More interestingly, the correlation between age and RR interval and FD parameters was apparently stronger in the control group than in the MVP group. Age is definitely an important factor that affects HRV. In addition, any factors that affect the autonomic nervous system, such as MVP in this study, will have an additive effect on HRV according to age. This suggests that patients with MVP may have autonomic nervous system involvement that increases the risk of arrhythmia and heart disease with increasing age.

The small sample size and differences in age distribution between the two groups are the limitations of this study. Furthermore, information on the environmental influences during 24 h ECG recordings is lacking; thus, some occult diseases that may influence autonomic function, such as ischemic heart disease or other diseases that increase risk by age, cannot be well identified in this study. Thus, further prospective studies are needed to gain more detailed information.

## 5. Conclusions

HRV declined significantly more in the MVP group than in the control group as age increased. Age would prospectively be associated with more of a decline in TD and FD of HRV measurements in some individuals, such as patients with MVP.

## Figures and Tables

**Figure 1 jcm-12-00165-f001:**
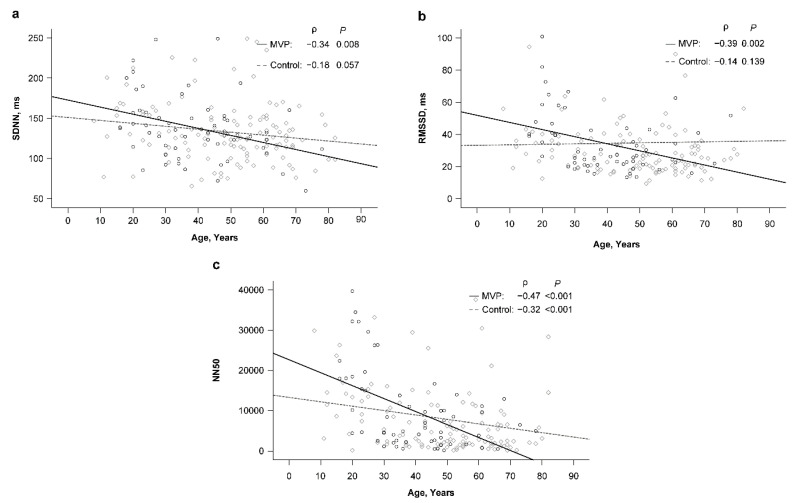
(**a**) Correlation between age and SDNN in the MVP and control groups. (**b**) Correlation between age and RMSSD in the MVP and control groups. (**c**) Correlation between age and NN50 in the MVP and control groups. The circle and diamond points indicate the MVP group and control group, respectively.

**Figure 2 jcm-12-00165-f002:**
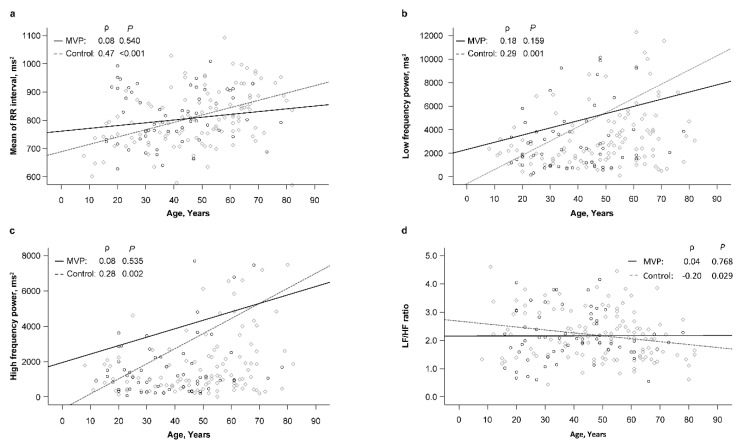
(**a**) Correlation between age and mean of RR interval in the MVP and control groups. (**b**) Correlation between age and low frequency power in the MVP and control groups. (**c**) Correlation between age and high-frequency power in the MVP and control groups. (**d**) Correlation between age and LF/HF ratio in the MVP and control groups. The circle and diamond points indicate the MVP group and control group, respectively.

**Table 1 jcm-12-00165-t001:** Basic demographics of patients with MVP and the control subjects.

	Before Propensity Score Matching			After Propensity Score Matching		
Variable	MVP (*n* = 60)	Control (*n* = 120)	*p*-Value	MVP (*n* = 52)	Control (*n* = 52)	*p*-Value
age, year	39.0 ± 15.4	48.9 ± 18.1	<0.001	41.5 ± 15.0	39.8 ± 18.5	0.621
female	46 (76.7)	80 (66.7)	0.227	41 (78.8)	40 (76.9)	1.000
heart rate, bpm	77.0 ± 8.1	77.1 ± 9.9	0.946	77.9 ± 7.9	79.4 ± 11.0	0.427
smoking	11 (18.3)	12 (10.0)	0.154	6 (11.5)	6 (11.5)	1.000
coffee	10 (16.7)	28 (23.3)	0.338	10 (19.2)	11 (21.2)	1.000
insomnia	6 (10.0)	10 (8.3)	0.783	5 (9.6)	5 (9.6)	1.000
syncope	6 (10.0)	18 (15.0)	0.486	5 (9.6)	7 (13.5)	0.760

Data are presented as frequency (%) or mean ± standard deviation.

**Table 2 jcm-12-00165-t002:** HRV parameters of the MVP group and the control group.

	Before Propensity Score Matching			After Propensity Score Matching		
Variable	MVP (*n* = 60)	Control (*n* = 120)	*p*-value	MVP (*n* = 52)	Control (*n* = 52)	*p*-Value
SDNN, ms	132.2 [113.1, 156.9]	127.3 [112.0, 153.4]	0.386	130.6 [111.7, 147.3]	130.2 [113.9, 158.8]	0.480
RMSSD, ms	27.3 [21.5, 42.6]	26.9 [19.1, 37.6]	0.215	26.3 [21.3, 39.5]	27.8 [20.1, 41.9]	0.616
NN50	6504 [2567, 14,964]	4230 [1722, 11,281]	0.077	4926 [2275, 11,003]	7363 [2103, 14,239]	0.384
PNN50, %	6.3 [2.3, 14.9]	4.5 [1.7, 11.4]	0.119	4.8 [2.1, 11.1]	6.4 [2.1, 12.9]	0.470
RR interval						
mean, ms^2^	796.1 [732.1, 905.4]	805.1 [738.2, 886.4]	0.813	785.8 [730.0, 874.8]	766.2 [735.2, 858.1]	0.833
VLF, ms^2^	2958 [1482, 4958]	2815 [1773, 4636]	0.844	2747 [1427, 4972]	2804 [1922, 4704]	0.644
LF, ms^2^	2402 [1191, 4655]	2729 [1540, 5096]	0.381	2448 [1191, 4724]	2361 [1540, 4681]	0.805
HF, ms^2^	1179 [522, 2613]	1168 [658, 2680]	0.603	1179 [522, 2787]	1046 [626, 2485]	0.969
Total, ms^2^	6449 [3238, 11,540]	7151 [4172, 12,800]	0.455	6449 [3238, 12,074]	6884 [4113, 12,061]	0.673
LF/HF ratio	2.05 [1.60, 2.80]	2.08 [1.46, 2.79]	0.999	2.05 [1.60, 2.86]	2.03 [1.42, 2.99]	0.828

Data are presented as median [25th, 75th percentile].

**Table 3 jcm-12-00165-t003:** Multivariable quantile regression for the association between MVP and HRV parameters using the whole cohort (*n* = 180).

Variable	Regression coefficient of MVP (95% CI) *	*p*-Value
SDNN, ms	3.6 (−8.6, 15.8)	0.560
RMSSD, ms	−1.5 (−7.3, 4.2)	0.602
NN50	38.3 (−2494.6, 2571.2)	0.976
PNN50, %	0.35 (−2.49, 3.19)	0.810
RR interval		
mean, ms^2^	5.6 (−36.2, 47.4)	0.792
VLF, ms^2^	246.1 (−749.4, 1241.5)	0.626
LF, ms^2^	−81.5 (−1128.5, 965.4)	0.878
HF, ms^2^	−79.1 (−713.7, 555.5)	0.806
Total, ms^2^	−229.5 (−2821.7, 2362.7)	0.861
LF/HF ratio	−0.10 (−0.51, 0.32)	0.653

* CI, confidence interval; adjusted for age, sex, and smoking.

**Table 4 jcm-12-00165-t004:** Correlation between age and HRV parameters in the MVP group and the control group.

	MVP		Control	
Variable	ρ	*p*-Value	ρ	*p*-Value
SDNN, ms	−0.34	0.008	−0.18	0.057
RMSSD, ms	−0.39	0.002	−0.14	0.139
NN50	−0.47	<0.001	−0.32	<0.001
PNN50, %	−0.46	<0.001	−0.24	0.008
RR interval				
mean, ms^2^	0.08	0.540	0.47	<0.001
VLF, ms2	0.09	0.500	0.17	0.062
LF, ms2	0.18	0.159	0.29	0.001
HF, ms2	0.08	0.535	0.28	0.002
Total, ms2	0.12	0.353	0.27	0.003
LF/HF ratio	0.04	0.768	−0.20	0.029

## Data Availability

The datasets generated during and/or analyzed during the current study are available from the corresponding author on reasonable request.

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
