# Peer review of "Effect of Age on Heart Rate Variability in Patients with Mitral Valve Prolapse: An Observational Study"

_jcm, 2022, doi:10.3390/jcm12010165_

Round 1

Reviewer 1 Report

The manuscript by Huang et coll. reports a study aiming to clarify the effect of age on heart rate variability in patients having mitral valve prolapse.

Here below my comments:

ABSTRACT

From line 17 the abstract is far to be rigorous. A time domain analysis is an analysis of physical signals in reference to time; frequency domain is an analysis of signals in reference to frequency. Which signal are you analyzing? I suppose RR sequence. If you do not indicate what you are analyzing this sentence does not have any sense. Again, stating that ‘…older patients had significantly lower time domain’ does not make any sense either. Same remark for the two subsequent sentences.  The last sentence is totally obscure to the reviewer. Please reformulate

INTRODUCTION

line 30-34: I imagine that, in this paragraph, you are referring to the fact that a subset of patients with symptomatic mitral valve prolapse also have associated autonomic dysfunction. The way these two sentences are written is highly confusing and can be source of misunderstands, please reformulate.

Line 36: ‘Evaluating HRV…’ does not provide ‘…cardiac electrophysiological signals’. Please correct

Line 45:  What do you mean for ‘long-term effects of HRV’?

Line 46: I do not understand your hypothesis. Please explain it better.

Line 48: again: what do you mean for ‘long-term effects of HRV’?

Overall, both abstract and introduction are really confusing and somewhere incorrect. Authors should consider to rewrite these two sections paying attention in being much more accurate and precise.

RESULTS

Line 124: ‘...older patients…’ , please specify the age after that you consider a patient as ‘old’.

Moreover, are the correlation coefficients reported in table 4 calculated considering all MVP patients or only ‘old’ patients? What about the control?

MINORS

Material and Methods:

line 77:  delete the bullet before ‘Equal….’

Line 124:  delete ‘…. lower TD,….’

Line 127:  add after FD the word ‘parameters’.

Line 130:  add after FD the word ‘parameters’.

Line 168 : add after TD the word ‘parameters’.

Line 170:  add after FD and also after TD the word ‘parameters’

Line 172 : add after FD the word ‘parameters’.

Please, increase the size of the legends of the axes in the figures. Generally, figures must be improved in quality. Please specify what does the circle represent and what does the diamond represent inside the diagrams.  

Author Response

Response to reviewer 1

ABSTRACT
From line 17 the abstract is far to be rigorous. A time domain analysis is an analysis of physical signals in reference to time; frequency domain is an analysis of signals in reference to frequency. Which signal are you analyzing? I suppose RR sequence. If you do not indicate what you are analyzing this sentence does not have any sense. Again, stating that ‘…older patients had significantly lower time domain’ does not make any sense either. Same remark for the two subsequent sentences. The last sentence is totally obscure to the reviewer. Please reformulate
Response: Thank you for your constructive comments. HRV declines as people get older. MVP has a low vagal tone. The MVP group seemed to have a different HRV trend as age increased compared with the control group in our study. We have revised the abstract accordingly in line 17 as per your suggestions.

INTRODUCTION
line 30-34: I imagine that, in this paragraph, you are referring to the fact that a subset of patients with symptomatic mitral valve prolapse also have associated autonomic dysfunction. The way these two sentences are written is highly confusing and can be source of misunderstands, please reformulate.
Response: We have revised this paragraph accordingly.

Line 36: ‘Evaluating HRV…’ does not provide ‘…cardiac electrophysiological signals’. Please correct
Response: We have deleted “providing cardiac electrophysiologic signals.”

Line 45: What do you mean for ‘long-term effects of HRV’?
Response: We apologize for this unclarity. Here we mean “longer periods” and not “long-term effects.” Hence, we have revised “long-term effects of HRV” to “the longer periods (24 h).”

Line 46: I do not understand your hypothesis. Please explain it better.
Response: We actually meant that HRV is affected by several factors, including MVP and that the effect of MVP on HRV is independent of age. It should be considered that 24-h period of ECG monitoring includes the periods of horizontal rest and vertical activity that have opposite effects on the autonomic system. This is also our study’s limitation. Thus, we tried to make our study group as simple as possible, with no influence of other factors. We analyzed HRV in the MVP and control groups and found the correlation of HRV with age. Further, we compared the HRV trend in these two groups. We have revised the hypothesis in the Introduction as follows: “HRV is affected by several factors, including MVP. We hypothesize that the effect of MVP on HRV is independent of age.”

Line 48: again: what do you mean for ‘long-term effects of HRV’?
Response: We have revised this to “24 h.”

Overall, both abstract and introduction are really confusing and somewhere incorrect. Authors should consider to rewrite these two sections paying attention in being much more accurate and precise.

RESULTS
Line 124: ‘...older patients…’ , please specify the age after that you consider a patient as ‘old’.
Moreover, are the correlation coefficients reported in table 4 calculated considering all MVP patients or only ‘old’ patients? What about the control?
Response: We want to mention the correlation between age and HRV parameters and not just for “old.” Table 4 depicts the calculated HRV of all patients with MVP and the controls. We have revised this issue in the subsection “3.3. Relationship between Age and HRV Parameters.”

MINORS
Material and Methods:
line 77: delete the bullet before ‘Equal….’ OK
Line 124: delete ‘…. lower TD,….’ OK
Line 127: add after FD the word ‘parameters’. OK
Line 130: add after FD the word ‘parameters’. OK
Line 168 : add after TD the word ‘parameters’. OK
Line 170: add after FD and also after TD the word ‘parameters’ OK
Line 172 : add after FD the word ‘parameters’. OK

Please, increase the size of the legends of the axes in the figures. Generally, figures must be improved in quality. Please specify what does the circle represent and what does the diamond represent inside the diagrams.
Response: Thanks for your constructive comments. We have enlarged the legends of the axes in the figures and increased the resolution of the figures. The circle points indicate the MVP group and the diamond points indicate the control group. We have expanded the legends of Figure 2.

Reviewer 2 Report

Huang and colleagues presented an interesting paper “Effect of age on heart rate variability in patients with mitral valve prolapse: An observational study”

However, I think some major revision must be done before publishing the paper.

First of all, the PVM population is significantly younger that the control population. A propensity match score should be performed in order to have two similar population.

Plus, no description of the mitral valve prolapse has be done. The degree of the prolapse, the co-existence of mitral valve regurgitation, the presence of Mitral Annular Disjunction should be taken in consideration if arrhythmic burden is taking into exam.

Were the patients asymptomatic?

Were they taking any drugs?

Plus, I think that a stress test such as Treadmill Test or Cardiopulmonary exercise test should be performed to analyse the chronotopic index in addition to HRV.

The topic is very interesting, some other exams should be performed and a similar control group should be used and the paper must be re-presented. 

Author Response

Response to reviewer 2

However, I think some major revision must be done before publishing the paper.

First of all, the PVM population is significantly younger that the control population. A propensity match score should be performed in order to have two similar population.

Response: Thank you for your insightful comment. We have created an additional propensity score-matched cohort to avoid potential confounding using the basic demographics listed in table 1 as the covariates from a multivariable logistic regression model. We have added the results of the matched cohort in Tables 1–2 and revealed consistent results. We have addressed this issue in subsections “2.6. Statistical Analysis,” “3.1. Patient Characteristics,” and “3.2 HRV Parameters between the MVP and Control Groups.”

Plus, no description of the mitral valve prolapse has be done. The degree of the prolapse, the co-existence of mitral valve regurgitation, the presence of Mitral Annular Disjunction should be taken in consideration if arrhythmic burden is taking into exam.

Response: We have added the description, inclusion, and exclusion criteria of the MVP in the subsection “2.2. Study population.”

Were the patients asymptomatic?

Response: All patients had palpitation and chest pain. We have revised subsection “2.2. Study population.”

Were they taking any drugs?

Response: The patients were not taking any drugs. We have rewritten the subsection “2.2. Study population,” accordingly.

Plus, I think that a stress test such as Treadmill Test or Cardiopulmonary exercise test should be performed to analyse the chronotopic index in addition to HRV.

Response: We agree with you. We only included the study populations with symptoms, history, physical examinations, and medical records. As stated in the Conclusions, this is a study limitation, wherein the possible influence on autonomic function, such as ischemic heart disease, cannot be identified; therefore, further prospective studies and stress tests are needed to gain more detailed information.

The topic is very interesting, some other exams should be performed and a similar control group should be used and the paper must be re-presented. 

Response: We agree with you. In this study, we should increase the case number in MVP group. Other exams, such as stress test, as mentioned above should be performed.The lack of information about the environmental influences during long-term ECG recordings is a limitation of our study. These all are to avoid potential confounding during HRV recording.

Reviewer 3 Report

The authors present an original study aimed at assessing the possible effect of mitral valve prolapse on HRV. Specifically, they performed 24h ECG monitoring and calculated HRV indices in 180 subjects (60 with MVP and 120 controls). The data analysis included study group comparison and multivariate quintile regression to find possible relationships between HRV indices and MVP, also pairwise Pearson correlations were assessed between HRV indices and age(?). No association between MVP and HRV indices was found. Pearson correlations between age and HRV indices were somewhat different in the study groups. Despite the lack of evidence for the relationships between HRV and MVP the authors nevertheless concluded that there is a dual effect of the MVP and age on the autonomic system.

Comments

1.       The formulation of the study hypothesis is hard to understand, “We hypothesized that HRV has an additive effect confounded by many other factors, such as MVP, besides age.” Additive effect on what? Probably, the authors actually meant that HRV is affected by many factors including MVP and that the effect of MVP on HRV is independent of age.

2.       The aim of the study is written as “to investigate the long-term effects of HRV on different age groups of patients with MVP”, however, the actual groups of the study are MVP and control group.

3.       The study population is not sufficiently defined, specifically, no inclusion criteria in the study groups are provided. Who was included in the MVP group (how accurately/reliably the MVP was detected), who was included in the control group? Age limits?

4.       The period, for which the HRV indices were calculated, is not indicated. It should be taken into account that 24-h period of ECG monitoring includes the periods of horizontal rest and vertical activity that have opposite effects on the autonomic system and thereby HRV indices so that 24-h indices are hard to interpret.

5.       The choice of the statistical methods must be based on the actual test of the assumption and not merely the idea that “The variation of TD and FD was large and may not satisfy the normality assumption”, which is not necessarily true. Unlike the t-test that in essence evaluates the difference in the sample means, i.e. the specific form of the sample difference - shift of the distribution, nonparametric analogs test the differences in means, variability, skewness, excess as a whole so that the significance of the test may be driven by any of these distributional moments or by all together without any clue on how much each of them contributes to that significance.

6.       The model(s) of multivariate relationships between MVP and HRV indices is not described, variables included in the model are not listed, the authors obviously suggest that age affects HRV, but no regression coefficient for this variable is provided.

7.       It is written ‘In the MVP group, older patients significantly correlated with lower TD, SDNN”, however, no numerical definition is provided for such subgroup

8.        The conclusion is quite confusing, “The dual effect of MVP and age causes a more significant effect on the autonomic nervous system as age increases…” More significant than what?

9.       This conclusion is not supported by the study data, the different relationships of HRV indices with age in MVP and control group by no means may indicate the relationships between MVP and HRV, in fact, it is only indicate the relationship between HRV and age reflecting the universal change of autonomic tone with ageing.

1.   Detection of the effect requires a prospective study with sufficient follow-up period between assessments of ‘cause’ and ‘effect’ variables. Cross-sectional study may only detect the associations, not causality

Author Response

Response to reviewer 3

Comments and Suggestions for Authors
The authors present an original study aimed at assessing the possible effect of mitral valve prolapse on HRV. Specifically, they performed 24h ECG monitoring and calculated HRV indices in 180 subjects (60 with MVP and 120 controls). The data analysis included study group comparison and multivariate quintile regression to find possible relationships between HRV indices and MVP, also pairwise Pearson correlations were assessed between HRV indices and age(?). No association between MVP and HRV indices was found. Pearson correlations between age and HRV indices were somewhat different in the study groups. Despite the lack of evidence for the relationships between HRV and MVP the authors nevertheless concluded that there is a dual effect of the MVP and age on the autonomic system.

Comments
The formulation of the study hypothesis is hard to understand, “We hypothesized that HRV has an additive effect confounded by many other factors, such as MVP, besides age.” Additive effect on what? Probably, the authors actually meant that HRV is affected by many factors including MVP and that the effect of MVP on HRV is independent of age.
Response: Thank you for highlighting this. We have revised this sentence to “HRV is affected by many factors, including MVP. We hypothesize that the effect of MVP on HRV is independent of age.” in the Introduction.

The aim of the study is written as “to investigate the long-term effects of HRV on different age groups of patients with MVP”, however, the actual groups of the study are MVP and control group.
Response: The original study aimed to assess the possible effect of MVP on HRV. Thus, the study design’s first step is to compare the MVP and control groups to discuss MVP as a factor of HRV. This issue is addressed in subsection “3.2 HRV Parameters between the MVP and Control Groups.” Then, we investigated the relationship between the patient’s age and HRV in the two groups. This issue is addressed in subsection “3.3 Relationship between Age and HRV Parameters.”

The study population is not sufficiently defined, specifically, no inclusion criteria in the study groups are provided. Who was included in the MVP group (how accurately/reliably the MVP was detected), who was included in the control group? Age limits?
Response: We have created an additional propensity score-matched cohort to avoid potential confounding using the basic demographics listed in table 1 as the covariates from a multivariable logistic regression model. We added the results of the matched cohort in Tables 1–2 and revealed consistent results. We have addressed this issue in subsections “2.6. Statistical Analysis,” “3.1. Patient Characteristics,” and “3.2 HRV Parameters between the MVP and Control Groups.” We also added the description and inclusion and exclusion criteria of the MVP in subsection “2.2. Study population.”

The period, for which the HRV indices were calculated, is not indicated. It should be taken into account that 24-h period of ECG monitoring includes the periods of horizontal rest and vertical activity that have opposite effects on the autonomic system and thereby HRV indices so that 24-h indices are hard to interpret.
Response: The frequency methods usually provide results that are more easily interpretable in terms of physiological regulations although the time domain methods, especially the SDNN and RMSSD methods, are used to investigate recordings of short durations. In general, the time domain methods are ideal for long-term recording analysis (the lower stability of heart rate modulations during long-term recordings makes the results of frequency methods less easily interpretable). The lack of information about the environmental influences during long-term ECG recordings is a limitation of our study. We have added this in the discussion and limitation.

The choice of the statistical methods must be based on the actual test of the assumption and not merely the idea that “The variation of TD and FD was large and may not satisfy the normality assumption”, which is not necessarily true. Unlike the t-test that in essence evaluates the difference in the sample means, i.e. the specific form of the sample difference - shift of the distribution, nonparametric analogs test the differences in means, variability, skewness, excess as a whole so that the significance of the test may be driven by any of these distributional moments or by all together without any clue on how much each of them contributes to that significance.
Response: We agree with you. The Kolmogorov-Smirnov normality tests on the HRV parameters revealed that none have met the normal distribution assumption. We have addressed this in subsection “2.6. Statistical Analysis.”

The model(s) of multivariate relationships between MVP and HRV indices is not described, variables included in the model are not listed, the authors obviously suggest that age affects HRV, but no regression coefficient for this variable is provided.
Response: The full results of the multivariable linear regression models are shown in Supplemental table 1. The results revealed a significant effect of age on the following HRV parameters: SDNN, RMSSD, NN50, PNN50, mean RR interval, LF, and total RR interval. We have addressed this in subsection “3.2 HRV Parameters between the MVP and Control Groups.”

It is written ‘In the MVP group, older patients significantly correlated with lower TD, SDNN”, however, no numerical definition is provided for such subgroup
Response: We simply want to emphasize the correlation between age and HRV parameters, and not just for “old.” Table 4 was calculated considering all patients with MVP and the controls. We have revised this issue in subsection “3.3. Relationship between Age and HRV Parameters.”

The conclusion is quite confusing, “The dual effect of MVP and age causes a more significant effect on the autonomic nervous system as age increases…” More significant than what?
Response: The conclusion should mention the correlation of HRV and age in MVP group compared with control group in our study. Thus, we revised that “HRV declines significantly in the MVP group than in the control group as the age increases,” and we have added this paragraph in the Conclusions.

This conclusion is not supported by the study data, the different relationships of HRV indices with age in MVP and control group by no means may indicate the relationships between MVP and HRV, in fact, it is only indicate the relationship between HRV and age reflecting the universal change of autonomic tone with ageing.
Response: We have created an additional propensity score-matched cohort to avoid potential confounding using the basic demographics listed in table 1 as the covariates from a multivariable logistic regression model. We added the results of the matched cohort in Tables 1–2 and revealed consistent results. We have addressed this issue in subsections “2.6. Statistical Analysis,” “3.1. Patient Characteristics,” and “3.2 HRV Parameters between the MVP and Control Groups.” As the title suggests, this observational study examines the effect of age on HRV in patients with MVP. It still has many topics to discuss, which is our limitation. Therefore, further prospective studies are needed to gain more information.

Detection of the effect requires a prospective study with sufficient follow-up period between assessments of ‘cause’ and ‘effect’ variables. Cross-sectional study may only detect the associations, not causality.
Response: This is our study limitation. Thus, we just analyzed the relationship between age and HRV on patients with MVP.

Round 2

Reviewer 1 Report

ABSTRACT

 line 18 : ‘No significant difference was found in the time domain and frequency domain between the two groups’ this sentence is incorrect.  A time domain analysis is an analysis of physical signals in reference to time. Frequency domain is an analysis of signals in reference to frequency, instead of time. So, what you are comparing in your analysis are several parameters calculated in time domain and other parameters calculated in frequency domain. You have to change this sentence in something like: ‘No significant difference was found in all parameters calculated neither in the time domain nor in the frequency domain between the two groups’

line 20: ‘as patients’ age increased, a significant time domain’. Again, this sentence doesn’t not make any sense. Which parameter/s calculated in time domain is/are changed according to patients’ age?  You should list them.

Line 22: ‘frequency domain were stronger in the control group. In 22 than in the MVP group’, same as before

Line 23: ‘….which had a significant decline in HRV as the increased’. This sentence is totally incomprehensible

Abstract has to be rewritten. In the present form it has several inaccuracies. As a whole it is totally meaningless as it is now.

INTRODUCTION

Line 49: longer of what? Please explain. Is HRV analysis normally calculated over period shorter than 24 h?  If that is the case you should explicit that in the text otherwise the sentence ‘the correlation of the longer periods (24 h) of HRV’ is without logic.

Line 54: ‘We hypothesize that HRV has an additive effect confounded by several other factors, such as MVP, besides age’ I do not understand this sentence, please reformulate it

MATERIALS AND METHODS

Line 102: ‘The variation of TD and FD was large’. This sentence is wrong: there is no variation neither in TD nor in TF. The variation is in the parameters calculated in time domain (SDNN, RMSSD, NN50 pNN50) and in parameters calculated in frequency domain (total power, VLF, LF, HF, LF/HF). Please change it.

Line 105: ‘The TD and FD between the MVP and control groups were compared using the Mann–Whitney U-test’ What you are comparing are the parameters calculated in the TD (or in the TF) not the TD and TF. Please modify.

RESULTS

Figures are in very low resolution. It is difficult to discriminate among circles and diamonds. Please increase the resolution of figures 1 and 2.  Please specify in Fig 1 legend  what does the circle represent and what does the diamond represent inside the diagrams. 

DISCUSSION:

Line 213: ‘during longer periods (24 h) ECG recordings is lacking’. Longer of what? Please explain

Author Response

Manuscript JCM-1915666

Response to Reviewers

Dear Reviewer

Thank you for giving us the opportunity to revise the manuscript “Effect of age on heart rate variability in patients with mitral valve prolapse: An observational study” for publication in the Journal of Clinical Medicine. We appreciate the time and effort that you and the reviewers dedicated to providing feedback on our manuscript and are grateful for the insightful comments on and valuable improvements to our paper. We have incorporated most of the suggestions made by the reviewers. Those changes are highlighted within the manuscript. Please see below, in blue, our point-by-point responses to the comments and concerns of the reviewers. All page numbers refer to the revised manuscript file with tracked changes.

Reviewers' Comments to the Authors:

ABSTRACT

 line 18 : ‘No significant difference was found in the time domain and frequency domain between the two groups’ this sentence is incorrect. A time domain analysis is an analysis of physical signals in reference to time. Frequency domain is an analysis of signals in reference to frequency, instead of time. So, what you are comparing in your analysis are several parameters calculated in time domain and other parameters calculated in frequency domain. You have to change this sentence in something like: ‘No significant difference was found in all parameters calculated neither in the time domain nor in the frequency domain between the two groups’

Author response: Thank you for raising this point. As suggested, we have revised the sentence as follows: “No significant difference was found in all parameters calculated in the time domain or in the frequency domain between the two groups.”

line 20: ‘as patients’ age increased, a significant time domain’. Again, this sentence doesn’t not make any sense. Which parameter/s calculated in time domain is/are changed according to patients’ age? You should list them.

Author response: Thank you for your constructive suggestion. We have listed SDNN, RMSSD, NN50, and pNN50 as changes according to patient’s age. For the time domain, the SDNN is the “gold standard” measure for medical stratification of cardiac risk when recorded over 24 h. Both the parasympathetic and sympathetic nervous systems contribute to SDNN. This trend occurs in our study, so we have emphasized this point in the abstract.

Line 22: ‘frequency domain were stronger in the control group. In 22 than in the MVP group’, same as before

Author response: Thank you for your constructive suggestion. We agree with you. We have deleted the sentence from the abstract.

Line 23: ‘….which had a significant decline in HRV as the increased’. This sentence is totally incomprehensible

Author response: Thank you for your comment. We agree with you. We wanted to state that in the MVP group, the involvement of the autonomic nervous system appears to be higher than that in the control group as age increases. Accordingly, we have rewritten the sentence in the abstract.

Abstract has to be rewritten. In the present form it has several inaccuracies. As a whole it is totally meaningless as it is now.

Author response: Thank you for your comment. We have rewritten the abstract accordingly.

INTRODUCTION

Line 49: longer of what? Please explain. Is HRV analysis normally calculated over period shorter than 24 h? If that is the case you should explicit that in the text otherwise the sentence ‘the correlation of the longer periods (24 h) of HRV’ is without logic.

Author response: We deleted the phrase “the longer periods (24 h) of.” We apologize for the misunderstanding. The European Society of Cardiology and the North American Society of Pacing and Electrophysiology created a task force recommending that the standardized measurement of HRV is either long-term (24 h) or short-term (5 min) recordings. Continuous 24-h HRV measurements represent the response of the cardiovascular system to a wide range of environmental stimuli and workloads. Circadian rhythms, core body temperature, metabolism, sleep cycle, and renin–angiotensin system contribute to the variability seen in 24-h HRV recordings. Short-term recordings are easier to conduct and involve fewer investigators/less participant burden and as such are most commonly used throughout nonclinical literature. However, these are not currently endorsed by the task force. In addition, as 24-h HRV measurements encompass measurements across the circadian rhythm, short-term values are not interchangeable with 24-h HRV values. Thus, we used 24-h ECG recording in the MVP group.

Line 54: ‘We hypothesize that HRV has an additive effect confounded by several other factors, such as MVP, besides age’ I do not understand this sentence, please reformulate it

Author response: Thank you for your constructive comments. We have deleted this sentence.

MATERIALS AND METHODS

Line 102: ‘The variation of TD and FD was large’. This sentence is wrong: there is no variation neither in TD nor in TF. The variation is in the parameters calculated in time domain (SDNN, RMSSD, NN50 pNN50) and in parameters calculated in frequency domain (total power, VLF, LF, HF, LF/HF). Please change it.

Author response: Thank you for the correction. We have made the required change.

Line 105: ‘The TD and FD between the MVP and control groups were compared using the Mann–Whitney U-test’ What you are comparing are the parameters calculated in the TD (or in the TF) not the TD and TF. Please modify.

Author response: Thank you for the correction; we have modified this sentence.

RESULTS

Figures are in very low resolution. It is difficult to discriminate among circles and diamonds. Please increase the resolution of figures 1 and 2. Please specify in Fig 1 legend what does the circle represent and what does the diamond represent inside the diagrams.

Author response: Thank you for your comment. We have increased the resolution of the figures. The circle and diamond points indicate the MVP group and control group, respectively. We have expanded the legends of Figure 1.

DISCUSSION:

Line 213: ‘during longer periods (24 h) ECG recordings is lacking’. Longer of what? Please explain

Author response: We apologize for the mistake. We have deleted “longer periods” and changed “long periods” to “long term.” The European Society of Cardiology and the North American Society of Pacing and Electrophysiology created a task force recommending that the standardized measurement of HRV is either long-term (24 h) or short-term (5 min) recordings. Continuous 24-h HRV measurements represent the response of the cardiovascular system to a wide range of environmental stimuli and workloads. Circadian rhythms, core body temperature, metabolism, sleep cycle, and renin–angiotensin system contribute to the variability seen in 24-h HRV recordings. Short-term recordings are easier to conduct and involve fewer investigators/less participant burden and as such are most commonly used throughout nonclinical literature. However, these are not currently endorsed by the task force. In addition, as 24-h HRV measurements encompass measurements across the circadian rhythm, short-term values are not interchangeable with 24-h HRV values. Thus, we used 24-h ECG recording in the MVP group.